# Validation of the Severity of Myalgic Encephalomyelitis/Chronic Fatigue Syndrome by Other Measures than History: Activity Bracelet, Cardiopulmonary Exercise Testing and a Validated Activity Questionnaire: SF-36

**DOI:** 10.3390/healthcare8030273

**Published:** 2020-08-14

**Authors:** C. (Linda) M. C. van Campen, Peter C. Rowe, Frans C. Visser

**Affiliations:** 1Stichting CardioZorg, 2132 HN Hoofddorp, The Netherlands; fransvisser@stichtingcardiozorg.nl; 2Department of Paediatrics, John Hopkins University School of Medicine, Baltimore, MD 21205, USA; prowe@jhmi.edu

**Keywords:** Sensewear^TM^ armband, chronic fatigue syndrome, cardiopulmonary exercise testing, peak VO_2_, VO_2_ at the ventilatory threshold, physical activity subscale, SF 36 questionnaire, disease severity, steps

## Abstract

Introduction: Myalgic encephalomyelitis/chronic fatigue syndrome (ME/CFS) is a severe and disabling chronic disease. Grading patient’s symptom and disease severity for comparison and therapeutic decision-making is necessary. Clinical grading that depends on patient self-report is subject to inter-individual variability. Having more objective measures to grade and confirm clinical grading would be desirable. Therefore, the aim of this study was to validate the clinical severity grading that has been proposed by the authors of the ME International Consensus Criteria (ICC) using more standardized measures like questionnaires, and objective measures such as physical activity tracking and cardiopulmonary exercise testing. Methods and results: The clinical database of a subspecialty ME/CFS clinic was searched for patients who had completed the SF 36 questionnaire, worn a Sensewear^TM^ armband for five days, and undergone a cardiopulmonary exercise test. Only patients who completed all three investigations within 3 months from each other—to improve the likelihood of stable disease—were included in the analysis. Two-hundred-eighty-nine patients were analyzed: 121 were graded as mild, 98 as moderate and 70 as having severe disease. The mean (SD) physical activity subscale of the SF-36 was 70 (11) for mild, 43 (8) for moderate and 15 (10) for severe ME/CFS patients. The mean (SD) number of steps per day was 8235 (1004) for mild, 5195 (1231) for moderate and 2031 (824) for severe disease. The mean (SD) percent predicted oxygen consumption at the ventilatory threshold was 47 (11)% for mild, 38 (7)% for moderate and 30 (7)% for severe disease. The percent peak oxygen consumption was 90 (14)% for mild, 64 (8)% for moderate and 48 (9)% for severe disease. All comparisons were *p* < 0.0001. Conclusion: This study confirms the validity of the ICC severity grading. Grading assigned by clinicians on the basis of patient self-report created groups that differed significantly on measures of activity using the SF-36 physical function subscale and objective measures of steps per day and exercise capacity. There was variability in function within severity grading groups, so grading based on self-report can be strengthened by the use of these supplementary measures.

## 1. Introduction

Chronic fatigue syndrome (CFS) is a potentially severe and disabling chronic disease [1,2,3]. The pathophysiology has not been established but there is considerable evidence that CFS is associated with multi-systemic neuropathology, metabolic, and immunological abnormalities [4,5,6,7,8,9,10,11,12,13,14,15,16]. In this light, the name “myalgic encephalomyelitis” was suggested by Carruthers and colleagues, a name also more consistent with the neurological classification of this disease in the World Health Organization’s International Classification of Diseases (ICD G93.3) [1].

Symptom and disease severity is discussed in the clinical application section of the International Consensus Criteria (ICC): “For a diagnosis of ME, symptom severity must result in a significant reduction in a patient’s premorbid activity level. Mild: approximately 50% reduction in activity, moderate: mostly housebound, severe: mostly bedbound, and very severe: bedbound and dependent on help for physical functions” [1]. One of the ICC’s considerations was to classify patient disease severity to increase patient group homogeneity in research. For example, we previously showed that the use of curcumin was only favorable in less severely ill ME/CFS patients and was not effective in severely ill ME/CFS patients [17].

History taking and clinical severity grading might be challenging because of its dependence on the interpretation of symptomatology by the patient. Patient expression of symptoms can be influenced by age, gender, education level, disease duration and co-morbidities like fibromyalgia, attention deficit disorder, and depression. Adding more objective measures to confirm symptom severity and clinical grading may be helpful. Severity grading is rarely reported in ME/CFS research and limited information is available on objective measures linked to disease severity. Also, the proposed grading of the ICC, which mainly focusses on the ability to perform physical activities, has not been validated.

Therefore, the aim of this study was to validate the clinical severity grading in ME/CFS, as suggested by the ICC [1], using both a standardized questionnaire (physical activity subscale of the SF-36 questionnaire) and two objective activity measures: a cardiopulmonary exercise test and a physical activity tracker.

## 2. Materials and Methods

### 2.1. Patients

From October 2012 to January 2018, 714 patients visited the outpatient clinic of the Stichting CardioZorg, Hoofddorp, the Netherlands, because of the suspicion of ME/CFS. This cardiology clinic specializes in diagnosing and treating adults with ME/CFS. All patients were evaluated by the same clinician (FVC). During the first visit, it was determined whether patients satisfied the criteria for CFS and ME, taking the exclusion criteria into account. Patients were classified as having chronic fatigue, or no chronic fatigue as defined by Fukuda and colleagues [3] and as having ME or no ME as defined by Carruthers and colleagues [1]. Disease severity was scored according to the ICC, with severity scored as mild, moderate, severe and very severe [1]. Very severe ME/CFS patients were not included in this analysis as none of these patients were able to undergo a cardiopulmonary exercise test (CPET).

Of the initial 714 patients, 675 were diagnosed with ME/CFS, fulfilling criteria of both ME and CFS. Of the remaining 39 patients, 26 did not fulfill the criteria of ME and/or CFS and 13 had an alternative diagnosis. The records of these 675 patients were searched for the availability of a completed SF-36 questionnaire, the availability of having worn a Sensewear^TM^ armband for five days and the availability of results of a cardiopulmonary exercise test. Only patients who completed all three investigations within a maximum interval of 3 months were included in the analysis. This interval was chosen to minimize variation in disease severity. Stability of disease was confirmed by review of the patient charts. Four hundred six patients had all three investigations. The interval for performance of the three measures exceeded 3 months in 75 patients, who were excluded. There were no differences in demographic data between ME/CFS patients who were included or excluded from the study due to the interval of the three measures (data not shown). Another 42 patients were excluded: because of a BMI > 37 (*n* = 7), because of gross under or overactivity compared to their average daily activity (*n* = 15), inadequate wearing armband time (*n* = 12), or motion artifacts (*n* = 8) (see below). We therefore included 289 patients in the analysis.

The study was carried out in accordance with the Declaration of Helsinki. All ME/CFS patients gave informed, written consent. The use of clinical data for descriptive studies was approved by the ethics committee of the Slotervaart Hospital (reference number P1450).

### 2.2. Sensewear^TM^ Activity Armband 

To track activity and help determine grading of disability, the Sensewear^TM^ armband (BodyMedia, Pittsburgh, PA, USA) was used. Patients wore the armband for approximately 5 days and were advised to take off the armband only during showering or bathing. Furthermore they were instructed to wear the armband if possible for a minimum of 23 h of the day. For assessing disability to guide treatment, patients were instructed to wear the armband in an on average reasonable period of their life, without excesses to be expected during the wearing time. From the armband data, the number of steps were recorded and normalized to 24 h to account for information of steps from only a part of the day. After wearing the armband patients were asked if the 5 days were “average” days. If there was a gross over- or under-activity due to mandatory physical activities or due to severe post-exertional malaise or current other illnesses (e.g., a viral infection), patients were excluded from the analysis. These exclusions were decided by the clinician and reported in the patient chart. Furthermore, patients who wore the armband less than 23 h. per day were excluded. Motion artifacts, due to horseback riding or motor bike riding, resulting in overestimation of steps, led to exclusion.

### 2.3. SF-36 Questionnaire: Physical Activity Subscale

SF-36 physical activity subscale asks whether the respondent’s health limits activities ranging from washing/clothing to walking shorter and longer distances and even strenuous running, performed during a typical day, ranging from limited a lot, limited a little, or not limited at all. The Dutch version of the SF-36 for physical activity [18] was used. The scores of the 10 items of the questionnaire were transformed into a scale ranging from 0–100%, where a higher score represents better physical condition and the lower scores worse conditions. Patients were instructed to complete questionnaires on average days, similar to the activity tracker, in order to avoid analysis on information acquired on a day with postexertional malaise or a good day.

### 2.4. Cardiopulmonary Exercise Testing

Patients underwent a symptom-limited maximal exercise test on a cycle ergometer (Excalibur, Lode, Groningen, The Netherlands) according to a previously described protocol [19]. A more detailed description can be found in Appendix A. 

### 2.5. Statistical Analysis

Data were analyzed using Graphpad Prism version 8.4.2 (Graphpad software, La Jolla, CA, USA) and using SPSS version 21 (IBM USA). All continuous data were tested for normal distribution using the D’Agostino–Pearson omnibus normality test. Data are presented as the mean (SD) or as median and interquartile range (IQR), where appropriate. Groups were compared using the paired or unpaired *t*-test where appropriate. Categorical and distribution data were tested by Chi-square analysis (3 × 2 table). Receiver operating curve (ROC) analysis was performed on the rand physical activity subscale of the SF-36, the number of steps on an activity meter, on the %VT VO_2_ of the cardiopulmonary exercise test and on the %peak VO_2_ of the cardiopulmonary exercise test to determine optimal cut-off values discriminating between mild and moderate and moderate and severe disease. For the analysis of the ROC curve Graphpad Prism was used. The sensitivities and specificities for different values of the different test measures were tabulated. This was performed separately for the mild versus moderate disease category as well as for the moderate versus severe disease category. This analysis resulted in an area under the curve (AUC), by a graphical representation of sensitivity% on the *y*-axis and 100%- specificity%. The most optimal discriminative value was obtained by the highest value of the multiplication of sensitivity with specificity [20,21]. Kappa’s with 95% confidence intervals were also calculated to determine agreement between clinical severity grading and the other measures as physical activity subscale of the SF-36, the number of steps per day, the %VT VO_2_ of the cardiopulmonary exercise test and on the %peak VO_2_ of the cardiopulmonary exercise test. The 95% confidence intervals (CI) of the Kappa values that overlap are considered equally discriminative. A Cohen’s kappa was calculated comparing the severity grading of two clinicians to determine reliability of clinical grading between the two. For this purpose a clinician (CMCvC) reviewed the charts of the first 162 patients. Within group comparison was done by the ordinary one way variance of analysis (ANOVA). Where significant, results were then explored further using the post-hoc Tukey’s test. Nominal data were compared using the Chi-square test (in a 3 × 2 table). Within group comparison with two different categorical independent variables on one continuous dependent variable was done by the two-way mixed analysis of variance (ANOVA). Where significant, results were then explored further using the post-hoc Tukey test. A *p*-value of <0.05 was considered to be statistically significant.

## 3. Results

### Patient Clinical Data

The studied group consisted of 289 ME/CFS patients and included 51 males (17.6%) and 238 females (82.4%). The mean (SD) age was 40 (11) years, the median BMI 23.4 (20.8–27.1) kg/m^2^ and the mean disease duration 12 (9) years. The mean physical activity subscale score from the SF-36 was 48 (24). The mean number of steps per day was 5701 (2670). The oxygen consumption at the ventilatory threshold was 11 (3) mL/kg/min, and the mean percentage predicted oxygen consumption at the ventilatory threshold was 40 (11)%. The peak oxygen consumption was 21 (7) mL/kg/min and the percentage predicted peak oxygen consumption was 71 (20)%. Using the clinician-assigned ICC severity category, 121 (42%) were scored as having mild disease, 98 patients (34%) were scored as having moderate disease and 70 patients (24%) were scored as having severe disease. The calculated Cohen’s kappa of the agreement in severity grading between the two clinicians was 0.86.

Table 1 shows the comparison of patient data between male and female ME/CFS patients. Only the peak VO_2_ data were significantly higher in male ME/CFS patients compared to female ME/CFS patients (*p* = 0.005).

Table 2 shows the baseline criteria and the results of the three tests for the different disease severity groups. Because the peak VO_2_ was significantly higher in male ME/CFS patients and percent VT VO_2_ and percent peak VO_2_ were not different between female and male ME/CFS patients, for the comparison of the severity grading, the percent VO_2_ data were used. No differences were found in baseline characteristics as age or disease duration. No significant difference was found between the ratio male/female patients. The physical activity subscale of the SF-36 was 70 (11) for mild, 43 (8) for moderate and 15 (10) for severe ME/CFS patients. The mean number of steps per day was 8235 (1004) for mild, 5195 (1231) for moderate and 2031 (824) for severe disease. The percent predicted oxygen consumption at the ventilatory threshold was 47 (11%) for mild, 38 (7%) for moderate and 30 (7%) for severe disease. The percent peak oxygen consumption was 90 (14%) for mild, 64 (8%) for moderate and 48 (9%) for severe disease. All comparisons were highly significantly different (all *p* < 0.0001). Figure 1 shows the graphical representation of those data. 

To compare the overall ICC severity grading with the other measures of activity, we divided the groups based on the results of Receiving Operator Characteristic (ROC) curves that generated the best cut-off values between mild and moderate and between moderate and severe disease. For the physical activity subscale of the SF-36, the best cut-off values were <30, from 30-to 60 and >60 to optimally discriminate between severe, moderate and mild disease. The area under the curve between moderate and severe was 0.984 and between mild and moderate 0.981. For the number of steps the best cut-off values were <3500, from 3500 to 6250 and >6250 steps to optimally discriminate between severe, moderate and mild disease. The area under the curve between moderate and severe was 0.997 and between mild and moderate 0.962. For the percent predicted oxygen consumption at the ventilatory threshold the best cut-off values were <30%, from 30% to 45% and >45%. The area under the curve between moderate and severe was 0.794 and between mild and moderate 0.760. For the percent predicted peak oxygen consumption the best cut-off value were ≤57%, from 58% to 72% and >72%. The area under the curve between moderate and severe was 0.925 and between mild and moderate 0.973. 

Table 3 shows the percentage of patients who are included in the range of the three predetermined cut-off values. Figure 2 shows the graphical representation of these results.

From the measures of agreement and confidence intervals, it follows that using the %VT VO_2_ for severity grading has the lowest kappa. Grading using the physical activity subscale of the SF-36, the number of steps per day and the %peak VO_2_ are equal with respect to the measured kappa, but the number of steps per day is superior to the %peak VO_2_ with respect to severity grading, as the 95% CI do not overlap.

Table 4 shows the results of a subgroup analysis: the objective results of female and male ME/CFS patients. As in the overall population, differences between mild, moderate and severe are all highly significantly different both in males and females. The 2-way ANOVA showed no significant differences in the severity groups and showed no significant interaction effect between the three disease severity groups and gender for the number of steps, the %VT VO_2_ and the %peak VO_2_. The physical activity subscale of the SF-36 showed a significant interaction between gender and disease severity (*p* = 0.03). Post hoc results are shown in the table.

## 4. Discussion

The ICC ME criteria proposed a severity classification including mild, moderate, severe and very severe disease [1]. The ICC’s considerations to classify patient disease severity were to increase patient group homogeneity in the research, to help orient and monitor treatment, and to determine total disease burden. Thus far, we are not aware of any studies that have implemented this classification, nor are we aware of validation studies. The main finding of this study is that the physical function subscale of the SF-36 questionnaire, and objective measures such as the number of steps per day on an activity meter and measures of percent oxygen consumption at ventilatory threshold and at peak exercise, showed a clear distinction between mild, moderate and severe ME/CFS patients. The physical activity subscale of the SF-36, the number of steps per day and oxygen consumption data all decreased significantly with increasing ICC severity as is shown in Figure 1. Secondly, in the present study, differences in the SF-36 physical activity subscale, in the number of steps and in the oxygen consumption at the ventilatory threshold and at peak exercise of mild, moderate and severe patients were comparable in female and male ME/CFS patients, indicating the validity of the severity grading in both women and men as is shown in Table 1. These data are in contrast to the study of Faro et al. who demonstrated in a large patient population that the clinical phenotype of male ME/CFS patients differed from that of female ME/CFS patients [22]. In the study of Faro et al., the physical activity subscales of the SF-36 were significantly higher in men than in women. However, the authors did not analyze the physical functioning subscale in relation to the clinical severity degree, as no subdivision was made in patients based on a clinical grading as in the present study.

While the four measures were significantly different between mild, moderate and severe disease, Table 3 and Figure 2 show discrepancies between the four measures. The oxygen consumption at the ventilatory threshold has only a fair agreement (low Kappa value) with the clinical grading and is therefore not likely to be helpful. The reasons for the low diagnostic capacity of the ventilatory threshold are unknown and need to be explored in future studies. In a previous study, we showed highly significant correlations between the physical functioning scale, the number of steps/day and the percent peak oxygen consumption in female ME/CFS patients [23]. Despite the highly significant correlations, a large variation between the three measures was found in that study. Activity is partially determined by age, race, menopausal status, educational level, body mass index, depressive symptoms, smoking, chronic medical conditions, and pain [24]. Furthermore, physical activity can also vary due to social circumstances (taking care of parents and children, marriage status), age, and previous physical fitness. The peak VO_2_ is influenced by genetics, gender, age, training status, exercise mode, bedrest, altitude, body composition, medication, the capacity of the respiratory and circulatory systems to take up and transport oxygen, and the capacity of the working muscles to receive and use oxygen. In ME/CFS patients the degree of fatigue/exhaustion, post-exertional malaise, underlying metabolic abnormalities, fibromyalgic pain, kinesiophobia and the use of medication may further influence physical activities. Moreover, the classification based on history taking may be difficult, due to a difference of interpretation between clinicians as the history can differ due to co-morbidities, cognitive dysfunction, gender, age, disease duration, social status, personality, dependence on social security funding and, importantly, on the presence or absence of post-exertional malaise during history taking. Thus, due to the large number of influencing factors, the cut-off values for the physical functioning scale, number of steps and peak VO_2_ in relation to the severity of the disease cannot be taken as absolute values, but merely support the proper severity grading. 

### 4.1. Physical Activity Questionnaires in Previous ME/CFS Studies:

A large number of studies have examined the validity of the SF-36 questionnaire, showing that the physical activity subscale discriminates between various diseases and healthy controls [25,26,27,28,29,30,31,32,33]. In ME/CFS patients Jason et al. reported the ability of the different subscales of the SF-36 questionnaire to discriminate CFS patients from healthy controls [34]. The authors found that the physical activity subscale was not very optimal to discriminate between patients and healthy controls, using an area-under-the-curve (AUC) cut-off value of >0.90 for optimal discrimination. In the community-based sample, the AUC of the physical activity scale was 0.84 and in the tertiary care sample 0.87. On the contrary, another study found an AUC for assessing decrease in the physical activity scale of 0.91 [35], suggesting that the use of the physical activity score is valid with an acceptable discriminative value. The current study did not compare patients with healthy controls, instead patients with mild, moderate and severe disease were compared. We found excellent areas under the curves of 0.981 for the distinction between mild and moderate disease and 0.984 for the distinction between moderate and severe disease. However, when interpreting results, care must be taken that completion of the questionnaire is not done in a period of post-exertional malaise, as underreporting can be expected, or in a period of revitalization where over reporting can be expected. 

### 4.2. Number of Steps in Previous ME/CFS Studies

The Sensewear^TM^ armband is a triaxial accelerometer, which measures steps, determines upright and lying position, characterizes sleep and estimates energy expenditure. Two studies have compared the number of steps of the Sensewear^TM^ armband to other activity trackers [36,37]. Both studies found a high mean absolute percentage error for the Sensewear^TM^ armband, relative to other activity trackers, it was considered to be in the middle range of reliability. The mean absolute percentage error (MAPE) reflects the accuracy of a device tested against a gold standard. Both studies used counted steps as the gold standard against measured steps from the device [36,37]. However, the study of Wahl et al. studied sporting conditions and not daily life circumstances. The study of An et al. studied daily life circumstances besides comparison of devices on a treadmill. Both studies showed the Sensewear^TM^ underestimated the number of steps. In the study of An et al., subjects did not wear the device for the whole 24 h, but only at documented activity periods. In the current study, patients wore the device for 5 complete days, which is a completely different study protocol, making results less comparable. Nevertheless, the Sensewear^TM^ device has successfully been used in patients with rheumatoid arthritis [38] in hemodialysis [39], and chronic obstructive pulmonary disease patients [40], spinal cord injury patients [41], in obese patients [42] and in children [43]. 

The exercise intolerance in CFS patients, using a variety of accelerometers, has been demonstrated in a number of studies [44,45,46,47,48,49,50]. None of these studies explored the discriminative value of the number of steps for symptom severity. Our data, with areas under the curve of the distinction between mild and moderate disease of 0.962 and of the distinction between moderate and severe disease of 0.997 suggests that activity tracking can be used as a diagnostic criterion (see Table 2 and Figure 1). However, as with the questionnaire, care must be taken that patients wear the tracker on average days. In fact, 15 patients were excluded from analysis due to under/over-activity.

### 4.3. Cardiopulmonary Exercise Test in Previous ME/CFS Studies

Finally, cardiopulmonary exercise testing is considered the most objective way to characterize exercise performance in ME/CFS patients. Multiple studies have shown that peak oxygen consumption is reduced in the majority of ME/CFS patients [51,52,53,54,55,56,57,58,59,60,61,62,63]. However, only one study determined the relation between the peak VO_2_ and accelerometer data in female ME/CFS patients: higher peak VO_2_ values were related to a higher physical activity time, physical activity energy expenditure, and a mean energy expenditure [46]. In line with the study of Ickmans et al., our study showed that in more severely affected ME/CFS patients, activity as expressed by the number of steps is associated with a lower percent predicted peak oxygen consumption as well as with the percent predicted oxygen consumption at the ventilatory threshold as is shown in Table 2 and Table 3. Studies have shown that cardiopulmonary exercise test values of males and females differ due to a variety of factors, including weight, height, total body fat, total muscle mass, hemoglobin, cardiac volumes, and lung volumes [64,65,66,67,68,69]. In the present study we also found a difference between male and female ME/CFS patients with respect to the absolute peak oxygen consumption as is shown in Table 1 However, when percent predicted values were used, no significant difference was found between male and female patients. This is due to the fact that results are normalized to a reference population in which reference values of males are higher of females. Whether the same cut-off values of the physical activity subscale of the SF-36, the number of steps per day and the percent parameters of the cardiopulmonary exercise test can be used for men and women to discriminate between mild, moderate and severe ME/CFS needs to be determined in a larger male patient sample.

### 4.4. Summary of Previous ICC Clinical Severity Category Grading 

We have explored the ICC disease grading and reported this in several recent papers. In 99 female patients, we correlated the physical activity subscale of the SF-36 with more objective measures as the number of steps per day on an activity tracker and the percent predicted peak oxygen consumption. Subgroup analysis of a RER over or under 1.1, the presence or absence of fibromyalgia, the use or non-use of pain-medication and a subdivision above and below a BMI of 30 had no impact on the correlations [23]. 

Furthermore, we reported on 82 female ME/CFS patients undergoing a 2-day CPET protocol, including 31 patients with mild disease, 31 patients with moderate disease and 20 patients with severe disease according to the ICC criteria. With increasing disease severity, cardiopulmonary exercise variables like oxygen consumption at the ventilatory threshold and peak exercise and the workload at the ventilatory threshold and at peak exercise declined significantly with increasing disease severity [70]. 

### 4.5. Limitations

This was a retrospective study taking data from patients with a maximum interval of 3 month between the three different measurements. This was a retrospective study and data can be used as a guideline for prospective studies. Ensuring a stable patient population was confirmed by checking of the patient charts: no major changes in symptomatology were found over this period of time in all patients. Only 289 out of 675 fulfilled all the requirements for analysis. The main reason for data acquisition in these patients were social security claims. In 269 patients, not all three methods were obtained. This may have led to inclusion bias. We did not include a control population. A prospective study is needed to evaluate the variability in measurements over time. Although the Sensewear^TM^ activity meter probably underestimates the number of steps per day, a clear distinction in the three patient severity groups was shown. In addition, the Sensewear^TM^ activity meter is not available anymore, but the present commercial actographs and smart watches have step measurements included. ME/CFS is a heterogeneous disease with complex phenotyping such as differences in onset, differences in symptom combinations and specific comorbidities. This heterogeneity might influence results/outcomes and needs to be studied in future.

## 5. Conclusions

Disease severity grading as suggested by Carruthers in the ICC on ME/CFS is validated by using questionnaires and more objective measures as the number of steps or cardiopulmonary exercise test parameters. We showed a well-defined difference between ICC severity categories and physical activity on the SF-36 questionnaire, as well as in the number of steps per day and the percentage predicted oxygen consumption at the ventilatory threshold and peak exercise. With this information, the history of the patients reported outcomes on the disease can be confirmed and be more comprehensible for the patient and his/her caretakers, treating physicians and authorities. Moreover, it increases patient group homogeneity in ME/CFS research. 

## Figures and Tables

**Figure 1 healthcare-08-00273-f001:**
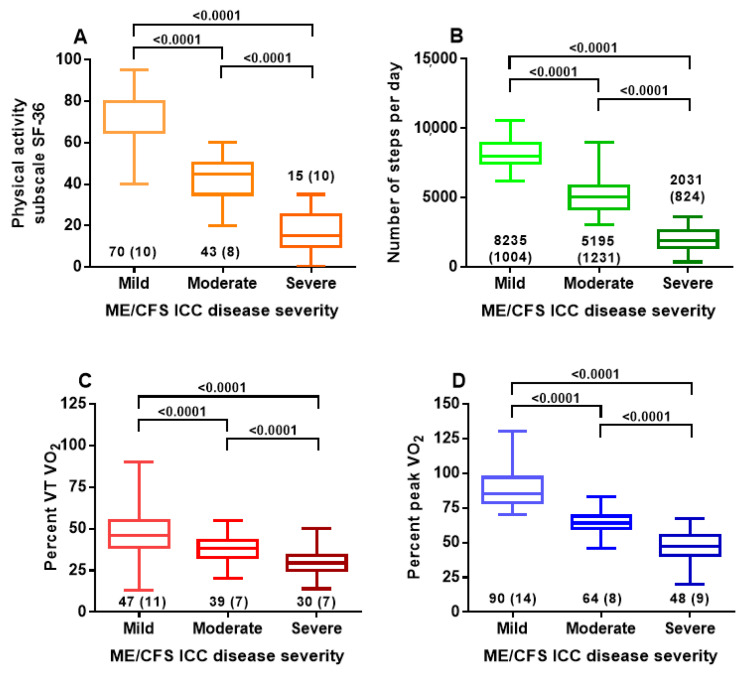
Physical activity subscale of SF-36 (panel **A**), number of steps per day (panel **B**), percent oxygen consumption at the ventilatory threshold (panel **C**) and percent peak oxygen consumption (panel **D**) in mild, moderate and severe ME/CFS according to clinical grading from International Consensus Criteria. Legend Figure 1 %peak VO_2_: oxygen consumption at peak exercise as percentile of a reference population; %VT VO_2_: oxygen consumption at the ventilatory threshold as percentile of a reference population.

**Figure 2 healthcare-08-00273-f002:**
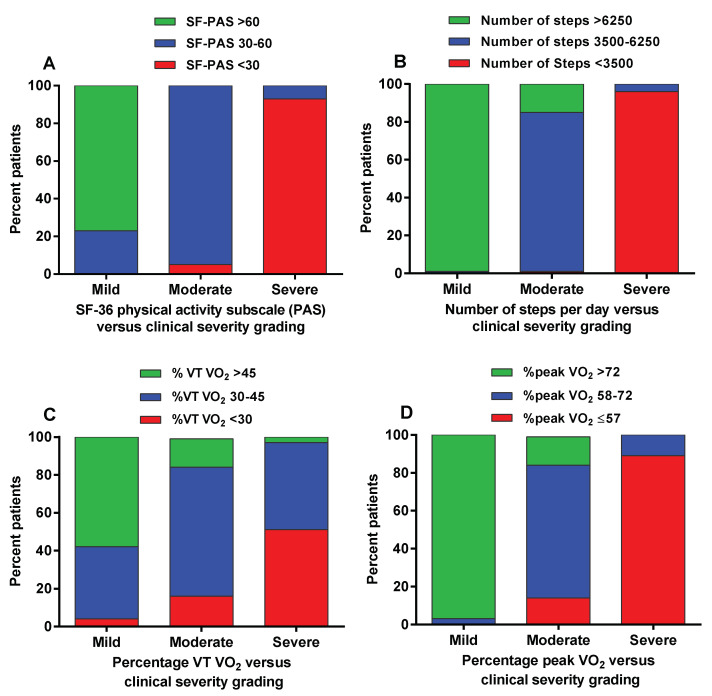
Stacked columns for physical activity subscale SF-36 (panel **A**) and number of steps per day (panel **B**), for percent oxygen consumption at the ventilatory threshold (panel **C**) and percent peak oxygen consumption (panel **D**) versus ICC clinical severity grading. Legend Figure 2 PAS: physical activity subscale of the SF-36 questionnaire; %peak VO_2_: oxygen consumption at peak exercise as percentile of a reference population; %VT VO_2_: oxygen consumption at the ventilatory threshold as percentile of a reference population.

**Table 1 healthcare-08-00273-t001:** Baseline characteristics all ME/CFS patients (*n* = 289) and for male (*n* = 51) and female ME/CFS patients (*n* = 238).

	Males (*n* = 51)	Females (*n* = 238)	*p*-Value
Age (years)	42 (11)	39 (11)	0.14
BMI (kg/m^2^)	24.9 (3.9)	24.5 (5.3)	0.64
Disease duration (years)	11 (8)	12 (9)	0.69
Disease severity: mild/moderate/severe (%) *	23/15/13 (45/29/26%)	98/83/57 (41/29/20%)	Chi-square 0.75
Heart rate at rest (bpm)	89 (16)	89 (19)	0.84
SBP at rest (mmHg)	125 (16)	130 (16)	0.18
DBP at rest (mmHg)	85 (20)	84 (11)	0.85
SF-36 PAS	48 (23)	48 (24)	0.96
Number of steps/day	5768 (2511)	5687 (2713)	0.84
VT VO_2_ (ml/kg/min)	12 (4)	11 (3)	0.09
%VT VO_2_	38 (14)	40 (11)	0.33
peak VO_2_ (ml/kg/min)	23 (8)	20 (6)	0.005
%peak VO_2_	72 (24)	71 (19)	0.79

BMI: body mass index; SBP: systolic blood pressure; DBP: diastolic blood pressure, HR: heart rate; PAS: physical activity subscale of the SF-36; peak VO_2_: oxygen consumption at peak exercise; %peak VO_2_: oxygen consumption at peak exercise as percentile of a reference population; VT VO_2_: oxygen consumption at the ventilatory threshold; %VT VO_2_: oxygen consumption at the ventilatory threshold as percentile of a reference population. * Disease severity was classified by the clinician using the ICC categories [1].

**Table 2 healthcare-08-00273-t002:** ME/CFS patient characteristics divided by ICC disease severity grading.

	Group 1: Mild	Group 2: Moderate	Group 3: Severe	One-Way ANOVA and Post-Hoc Tukey’s Test
Number	121	98	70	
Male/female	23/98	15/83	13/57	Chi-square 0.75 (3 × 2 table)
Age (years)	43 (11)	38 (11)	35(11)	F (2, 286) = 12.1; *p* < 0.0001. Post-hoc tests: 1 vs. 2 *p* = 0.008; 1 vs. 3 *p* < 0.0001 and 2 vs. 3 *p* = 0.12
Disease duration (years)	12 (9)	12 (8)	12 (8)	F (2, 286) = 0.06; *p* = 0.94
SF-36 PAS	70 (11)	43 (8)	15 (10)	F (2, 286) = 717.6; *p* < 0.0001. Post-hoc tests: 1 vs. 2 *p* < 0.0001; 1 vs. 3 *p* < 0.0001 and 2 vs. 3 *p* < 0.0001
Number of steps/day	8235 (1004)	5195 (1231)	2031 (824)	F (2, 286) = 792.4; *p* < 0.0001. Post-hoc tests: 1 vs. 2 *p* < 0.0001; 1 vs. 3 *p* < 0.0001 and 2 vs. 3 *p* < 0.0001
%VT VO_2_	47 (11)	38 (7)	30 (7)	F (2, 286) = 85.7; *p* < 0.0001. Post-hoc tests: 1 vs. 2 *p* < 0.0001; 1 vs. 3 *p* < 0.0001 and 2 vs. 3 *p* < 0.0001
%peak VO_2_	90 (14)	64 (8)	48 (9)	F (2, 286) = 355.0; *p* < 0.0001. Post-hoc tests: 1 vs. 2 *p* < 0.0001; 1 vs. 3 *p* < 0.0001 and 2 vs. 3 *p* < 0.0001

BMI: body mass index; PAS: physical activity subscale of the SF-36; %peak VO_2_: oxygen consumption at peak exercise as percentile of a reference population; %VT VO_2_: oxygen consumption at the ventilatory threshold as percentile of a reference population.

**Table 3 healthcare-08-00273-t003:** Subdivision of cut-off values/measures of agreement as determined by ROC analysis for the four studied measures in relation to ICC clinical severity grading.

	Group 1 Mild	Group 2 Moderate	Group 3 Severe	Measure of Agreement (Kappa)
Physical Activity subscale of the SF-36 (PAS) (panel A Figure 2)				
>60	93 (77%)	0 (0%)	0 (0%)	Kappa 0.80 (95%CI: 0.742–0.859)
30–60	28 (23%)	93 (95%)	5 (7%)	
<30	0 (0%)	5 (5%)	65 (93%)	
Number of steps (panel B Figure 2)				
>6250	120 (99%)	15 (15%)	0 (0%)	Kappa 0.89 (95% CI: 0.848–0.938).
3500–6250	1 (1%)	82 (84%)	3 (4%)	
<3500	0 (0%)	1 (1%)	67 (96%)	
%predicted oxygen consumption at the ventilatory threshold (panel C Figure 2)				
>56%	70 (58%)	15 (15%)	2 (3%)	Kappa 0.39 (95% CI: 0.303–0.473).
30–45%	46 (38%)	68 (69%)	32 (46%)	
<30%	5 (4%)	15 (16%)	36 (51%)	
%predicted peak oxygen consumption (panel D Figure 2)				
>72%	117 (97%)	15 (15%)	0 (0%)	Kappa 0.78 (95% CI: 0.721–0.843).
58–72%	4 (3%)	70 (71%)	8 (11%)	
≤57%	0 (0%)	13 (14%)	62 (89%)	

CI: confidence intervals.

**Table 4 healthcare-08-00273-t004:** Comparison of female and male ME/CFS patients on validated questionnaire and more objective measures.

	Group 1: Mild	Group 2: Moderate	Group 3: Severe	Two-Way Mixed ANOVA with Post Hoc Tukey Test
Female/Male ME/CFS patients				
Number of males/females	23/98	15/83	13/57	
Male SF-36 PAS	67 (13)	43 (7)	20 (11)	F (2, 283) = 3.55; *p* = 0.030. Post-hoc tests: female patients 1 vs. 2 *p* < 0.0001; 1 vs. 3 *p* < 0.0001 and 2 vs. 3 *p* < 0.0001; male patients 1 vs. 2 *p* < 0.0001; 1 vs. 3 *p* < 0.0001 and 2 vs. 3 *p* < 0.0001
Female SF-36 PAS	71 (10)	44 (9)	14 (10)	
Male Number of steps/day	8067 (1152)	5056 (892)	2523 (860)	F (2, 283) = 2.51; *p* = 0.083
Female Number of steps/day	8274 (969)	5220 (1286)	1919 (780)	
Male %VT VO_2_	48 (15)	33 (5)	26 (6)	F (2, 283) = 2.11; *p* = 0.12.
Female %VT VO_2_	47 (10)	39 (7)	30 (8)	
Male %peak VO_2_	93 (15)	60 (9)	46 (9)	F (2, 283) = 2.46; *p* = 0.088
Female %peak VO_2_	89 (13)	65 (7)	48 (9)	

PAS: physical activity subscale of the SF-36; %peak VO_2_: oxygen consumption at peak exercise as percentile of a reference population; %VT VO_2_: oxygen consumption at the ventilatory threshold as a percentile of a reference population.

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
