# Peer review of "Validation of the Severity of Myalgic Encephalomyelitis/Chronic Fatigue Syndrome by Other Measures than History: Activity Bracelet, Cardiopulmonary Exercise Testing and a Validated Activity Questionnaire: SF-36"

_healthcare, 2020, doi:10.3390/healthcare8030273_

Round 1

Reviewer 1 Report

In this manuscript, the authors are working to integrate and cross-validate various tools applied to the ME/CFS disease with the goal to improve/streamline the diagnostic process.

It seems the statistical analysis is sound even though I would not be able to reproduce such analysis without assistance. So this area of expertise will be deferred to another reviewer hopefully.

At first, the tables and figures seemed informative and the figures are certainly doing a good job at displaying the underlying analysis.

The following comments are the description of how the manuscript should be improved before publication. There are in the order of the text as no line number was provided in the pdf.

Results:

In my opinion, the results section is too descriptive and redundant. There has to be a better way to describe the data than just repeating in writing what is already in the tables and figures.

I will take the first paragraph as an example. Table 1 is merely laid down as text and parenthesis. Moreover, it is confusing because the text is not following the order of the table. Table 1 does not specify what the numbers are, and even though that information is in the text, some of the numbers are means and SD while other are proportions in numbers and %. I have been taught that a table should be self-sufficient and the text should stir the reader to what they deem relevant in that table.

Talking about relevance, Table 1 does not bring much to the paper, especially in light of Table 2. Tables 1 and 2 should be combined in 1 table with all the information from both tables (5 columns and 14 rows?).

The order male/female is opposite in Tables 1 and 3 vs other tables. So confusing to follow. You need consistency.

Can you explain why the fact that there is a significance difference in peak VO2 between males and females directs you to use the percent peak VO2 instead? It might be warranted to have this explanation in the manuscript.

The text states 70 (10) when Table 3 states 70 (11) and Figure 1 states 70 (10). Abstract states 70 (10). The text states 39 (7) when Table 3 states 38 (7) and Figure 1 states 38 (7). Abstract states 39 (7). Here again, such level of redundancy is pointless. I personally prefer figures over tables and those are well done figures. If you want to keep Table 3 because of the few characteristics not shown in Figure 1, you can move it to supplemental, or display all characteristics of Table 3 in an expanded Figure 1. Sentences like “No significant difference was found between the ratio male/female patients.” are far more useful than repeating the numbers from Table 3 and Figure 1.

Another example of redundancy, when you repeat the legend of Figure 1 in the text.

Why is there no significant detail about the ROC analysis and how it led to specific cut-offs other than just giving the results? This is an important part missing from the results in my opinion. A figure or table seems warranted here as the authors use those thresholds for Figure 2. Other researchers from the field might want to adopt those thresholds but won’t be able to without backed up from the analysis in this paper. Also, here, a table would be more useful than throwing a bunch of numbers in the text.

Page 7 top is almost exactly the same as legend of Figure 2.

Figure 2B should use a different abbreviation for number. It currently reads as “No steps”.

The description of Figure 2 is way too lengthy, redundant and brings little to the manuscript. This is the only relevant part in my opinion: “From the measures of agreement and confidence intervals, it follows that using the %VT VO2 for severity grading is insufficient. Grading using the physical activity subscale of the SF-36 ,the number of steps per day and the %peak VO2 are equal, but the number of steps per day is superior to the %peak VO2 with respect to severity.” It might be good to expand a bit but certainly not write down all the numbers as done above that paragraph.

I think Table 4 should be presented as a figure.

The way I see it, the tables are currently built in a way that Table 1 is a summary. Table 2 expands Table 1 a little bit. Table 3 expands Table 2 and Table 4 expands Table 3. There has to be a better way to use publication space without repeating the same numbers over and over.

Discussion:

The discussion has one reference to Figure 2. References to other tables and Figure 1 are very much missed when reading it. One of many examples where it would be useful is after this statement: “The oxygen consumption at the ventilatory threshold has only a fair agreement (low Kappa value) with the clinical grading and is therefore not likely to be helpful.”

Most of your discussion reads as a review with little reference to your result section.

This sentence: “Thus far, we are not aware of many studies that have implemented this classification, nor are we aware of validation studies.” is missing references or “many” should be “any”.

After (24), it should be “In that study”.

During the comparison with the Faro study, you state that you did not find differences between males and females in your analysis. You then state that they did find significant differences in the physical activity subscale which I assume you are referring to functioning and role functioning. PAS in not mentioned by Faro. I assume you are talking about your Table 2 and their Figure 2?

I am afraid I do not follow the logic of the following sentence: “However, they did not analyze the physical functioning subscale in relation to the clinical severity degree.” I don’t know which table or figure you are referring to in your manuscript. Where did you do that? What is in contrast with what between the 2 studies? Are you predicting that had they done that analysis, they would have found the same result as you? Please explain?

(25) should be cited after first sentence referring to it.

Several references are needed when a statement like “a large number of studies” have examined…” Not just one from 1993.

An et al. and Wahl et al. are not in the references.

What is a “high mean absolute percentage error”? Does it mean it overestimates steps?

“a activPAL” should be “an activPAL”.

After (36), it should be “In line with that study”.

There is a problem throughout the manuscript with abbreviations. Some are missing their meaning, like ADD, COPD. Others are not explained at first mention, like ICC in abstract. Others are defined correctly but then towards the end of the manuscript, they are not used anymore and instead the whole words are used again. It makes it hard to follow. ROC and CPET are good examples when both the abbreviations and the full meaning are used in different parts of manuscript but not defined correctly at first instance. VO2 is also not written with subscript at several spots (e.g. in Table 1). TM in abstract should be a superscript. PAS in Table 3 is not defined.

In methods, “let to” should be “led to”?

Author Response

In this manuscript, the authors are working to integrate and cross-validate various tools applied to the ME/CFS disease with the goal to improve/streamline the diagnostic process.

It seems the statistical analysis is sound even though I would not be able to reproduce such analysis without assistance. So this area of expertise will be deferred to another reviewer hopefully.

At first, the tables and figures seemed informative and the figures are certainly doing a good job at displaying the underlying analysis. Thank you

The following comments are the description of how the manuscript should be improved before publication. There are in the order of the text as no line number was provided in the pdf.

Results:

In my opinion, the results section is too descriptive and redundant. There has to be a better way to describe the data than just repeating in writing what is already in the tables and figures.

I will take the first paragraph as an example. Table 1 is merely laid down as text and parenthesis. Moreover, it is confusing because the text is not following the order of the table. Table 1 does not specify what the numbers are, and even though that information is in the text, some of the numbers are means and SD while other are proportions in numbers and %. I have been taught that a table should be self-sufficient and the text should stir the reader to what they deem relevant in that table. The data from the original Table 1 is now mentioned only in the text and the original Table 1 has been replaced by the new Table 1 describing differences between males and females.

 Talking about relevance, Table 1 does not bring much to the paper, especially in light of Table 2. Tables 1 and 2 should be combined in 1 table with all the information from both tables (5 columns and 14 rows?).

The order male/female is opposite in Tables 1 and 3 vs other tables. So confusing to follow. You need consistency. As suggested we have now included the original Table 1 information in the text and changed the original Table 2 into 1. We have made the other Tables more consistent.

 Can you explain why the fact that there is a significance difference in peak VO2 between males and females directs you to use the percent peak VO2 instead? It might be warranted to have this explanation in the manuscript. A comment regarding the difference in absolute oxygen consumption values between males and females was made in new lines 348 and following. In line with previous studies, peak oxygen consumption is higher in males compared to females. Due to normalizing test results as a percentage of a reference population, percent predicted results do not differ between males and females, as we have shown in this and several previous studies.

The text states 70 (10) when Table 3 states 70 (11) and Figure 1 states 70 (10). Abstract states 70 (10). The text states 39 (7) when Table 3 states 38 (7) and Figure 1 states 38 (7). Abstract states 39 (7). Here again, such level of redundancy is pointless. I personally prefer figures over tables and those are well done figures. If you want to keep Table 3 because of the few characteristics not shown in Figure 1, you can move it to supplemental, or display all characteristics of Table 3 in an expanded Figure 1. Sentences like “No significant difference was found between the ratio male/female patients.” are far more useful than repeating the numbers from Table 3 and Figure 1. Checked and corrected. We respectfully disagree with the reviewer, as the Table adds more information on statistics than is possible in a figure or text description. Moreover, reviewers of our other studies have stressed the importance of tables and statistics over figures. However, if the manuscript is accepted the guest editor may decide on this issue.

Another example of redundancy, when you repeat the legend of Figure 1 in the text. Deleted at your request.

Why is there no significant detail about the ROC analysis and how it led to specific cut-offs other than just giving the results? This is an important part missing from the results in my opinion. A figure or table seems warranted here as the authors use those thresholds for Figure 2. Other researchers from the field might want to adopt those thresholds but won’t be able to without backed up from the analysis in this paper. Also, here, a table would be more useful than throwing a bunch of numbers in the text. Not to flood the manuscript with numbers, we performed the ROC analysis as described by Liu and determined the cut-off values, taking the clinical grading in this case as “gold” standard: we showed the cut-off values with the highest number resulting from multiplying sensitivity with specificity, also resulting in the highest area under the curve. We respectfully disagree with the reviewer that showing more numbers in tables on cut-off values with lower results and lower values for area under the curve would not improve the manuscript.

Page 7 top is almost exactly the same as legend of Figure 2.altered

Figure 2B should use a different abbreviation for number. It currently reads as “No steps”. altered

The description of Figure 2 is way too lengthy, redundant and brings little to the manuscript. This is the only relevant part in my opinion: “From the measures of agreement and confidence intervals, it follows that using the %VT VO2 for severity grading is insufficient. Grading using the physical activity subscale of the SF-36 ,the number of steps per day and the %peak VO2 are equal, but the number of steps per day is superior to the %peak VO2 with respect to severity.” It might be good to expand a bit but certainly not write down all the numbers as done above that paragraph. We showed the results in a table for clarity, as suggested.

I think Table 4 should be presented as a figure. See below

The way I see it, the tables are currently built in a way that Table 1 is a summary. Table 2 expands Table 1 a little bit. Table 3 expands Table 2 and Table 4 expands Table 3. There has to be a better way to use publication space without repeating the same numbers over and over. We can understand this impression: the Table 1/2 matter has been resolved. The new Table 2 (old table 3) and Figure 1 belong together (where some readers prefer tables to figures and vice versa). The new Table 2 mainly has to stay for statistical simplification purposes. Describing this in the text would definitely not improve readability. Table 4 is too complex with the two-way statistical analysis and having males AND females to be shown in a figure.

Discussion:

The discussion has one reference to Figure 2. References to other tables and Figure 1 are very much missed when reading it. One of many examples where it would be useful is after this statement: “The oxygen consumption at the ventilatory threshold has only a fair agreement (low Kappa value) with the clinical grading and is therefore not likely to be helpful. Checked and updated where appropriate.

Most of your discussion reads as a review with little reference to your result section. The first two paragraphs of the discussion now summarize our study findings. The rest of the Discussion relates those results to what is known in literature. We incorporated our findings in the section on the physical activity questionnaire and the number of steps.

This sentence: “Thus far, we are not aware of many studies that have implemented this classification, nor are we aware of validation studies.” is missing references or “many” should be “any”. altered

After (24), it should be “In that study”. rephrased

During the comparison with the Faro study, you state that you did not find differences between males and females in your analysis. You then state that they did find significant differences in the physical activity subscale which I assume you are referring to functioning and role functioning. PAS in not mentioned by Faro. I assume you are talking about your Table 2 and their Figure 2? PAS as an abbreviation of physical functioning subscale in our study is the same as physical function in Faro’s study in figure 2 (the one most on the left). In the new Table 1, no difference between men and women on the physical functioning subscale was found in our study.

I am afraid I do not follow the logic of the following sentence: “However, they did not analyze the physical functioning subscale in relation to the clinical severity degree.” I don’t know which table or figure you are referring to in your manuscript. Where did you do that? What is in contrast with what between the 2 studies? Are you predicting that had they done that analysis, they would have found the same result as you? Please explain? We related the PAS and the number of steps and percentage oxygen consumption at the ventilatory threshold and at peak exercise with the clinical ICC grades, mild, moderate and severe and showed significant differences in all three measures. The study of Faro did not relate any described outcomes to clinical severity. We have clarified this with an additional comment.

(25) should be cited after first sentence referring to it. Dugan is cited after the first sentence. The study referred to in the sentence earlier is a different one and citing of that paper has been added.

Several references are needed when a statement like “a large number of studies” have examined…” Not just one from 1993. We have now added reference to meta-analyses, which in turn provide more detail on the large number of studies described in the manuscript.

An et al. and Wahl et al. are not in the references. Added.

What is a “high mean absolute percentage error”? Does it mean it overestimates steps? We added additional sentences to clarify.

“a activPAL” should be “an activPAL”. We deleted the comment of this reference, because we felt it was less relevant to the discussion.

After (36), it should be “In line with that study”. rephrased

There is a problem throughout the manuscript with abbreviations. Some are missing their meaning, like ADD, COPD. Others are not explained at first mention, like ICC in abstract. Others are defined correctly but then towards the end of the manuscript, they are not used anymore and instead the whole words are used again. It makes it hard to follow. ROC and CPET are good examples when both the abbreviations and the full meaning are used in different parts of manuscript but not defined correctly at first instance. VO2 is also not written with subscript at several spots (e.g. in Table 1). TM in abstract should be a superscript. PAS in Table 3 is not defined. Abbreviations, sub- and superscript checked and updated. Abbreviations were used less in the text to ensure better readability except for obvious ones like ICC. PAS was used in tables and explained in the footer of the table.

In methods, “let to” should be “led to”? altered

Reviewer 2 Report

The authors use retrospective analysis of data from one clinic collected over 6 years and including 289 patients diagnosed as ME/CFS using the 1994 CFS research case definition or the 2011 ME ICC criteria. The compared results of SF-36 physical activity subscale, steps measured with Sensewear activity armband, and a symptom-limited cardiopulmonary exercise tests with clinical classification of disease severity (mild, moderate, severe, very severe) according to ME ICC description. They conclude that the standardized and objective measures are significantly different by disease severity, and that this validates the ICC severity classification. The data are of interest, but the authors are asked to provide additional information and to re-frame this retrospective analysis as a pilot study evaluating different approaches to determining disease severity. Validation is a bit too definitive.
1.The study population is not fully defined. For clinical purposes, are patients classified as either ME or CFS and then treated/managed the same? Of the 714 presenting as presumed ME/CFS, what were the reasons that 39 were not considered to be ME/CFS (symptoms, exclusionary conditions, alternative diagnoses, other)? Other parameters of the complex ME/CFS phenotype are omitted, such as pain, sleep, cognition, orthostatic intolerance, PEM, medication use, sudden vs gradual onset, comorbidities use as fibromyalgia, etc. are not described. This information may not be available in retrospective review, but this kind of heterogeneity deserves comment and should be included as a limitation.
2. How do the patients with valid assessments for all 3 measures within 3 months compare with those who could not be included in the study? Of 675 patients with ME/CFS less than half [only 289 (40.5%)] are in the study. This could introduce significant bias in the assessment.
3. The inter-observer agreement for the disease severity classification of patients using the ICC criteria appears to quite good (kappa = 0.86), however this was performed on only slightly more than half of the study participants (162/289, 56%) and the second clinician made the determination through record rather than during clinic visit. Was the clinic physician’s determination of disease severity included in patient charts, if so, how was this redacted to allow unbiased determination. What is the reliability of clinical record determination of disease severity compared with determinations made at clinical visits?
4. The CPET is described as symptom limited and different rates of ramping are used. This does not seem to follow usual practice, where determinations are made using either maximal or submaximal challenge. This form of CPET could be useful, but further description and justification would be helpful. CPET can be stressful to patients with ME/CFS. It would be helpful for the authors to comment on any adverse outcomes associated with CPET.
5. The Sensewear activity armband step count was normalized to 24 hours. Why wasn’t the step count used directly? What is gained by normalizing to 24 hours?
6. The ICC proposal for disease severity includes 4 categories, but the report describes only 3. The very severe category may not be represented in the clinic population. In addition, their illness may preclude having all three measures, particularly the CPET. It would be helpful if this could be addressed by the authors.
7. Results shown in Figure 1 as a bar graph may be more informative if shown as box plots to give a clearer idea of the distribution and numbers. Data shown in Figure 2 may be more helpful shown in table form with sensitivity and specificity for each classification. The ROC data may be clearer if presented graphically on sensitivity/specificity analysis.
8. Discussion (section 4.4) presents information not on the study population not included in the results. The ambiguous use of “this study” should be avoided as it is confusing whether “this study” refers to the study being presented or the one referenced. What is being conveyed by the statement “Both studies found high mean absolute % error for Sensewear armband relative to other activity trackers”? This seems to suggest Sensewear may not be valid or reliable and some clarification is needed.

9. Additional limitations should be included. Only one clinical practice is evaluated. The data could be used to focus a prospective analysis of the most promising measures in multiple clinical settings.

Author Response

The authors use retrospective analysis of data from one clinic collected over 6 years and including 289 patients diagnosed as ME/CFS using the 1994 CFS research case definition or the 2011 ME ICC criteria. The compared results of SF-36 physical activity subscale, steps measured with Sensewear activity armband, and a symptom-limited cardiopulmonary exercise tests with clinical classification of disease severity (mild, moderate, severe, very severe) according to ME ICC description. They conclude that the standardized and objective measures are significantly different by disease severity, and that this validates the ICC severity classification. The data are of interest, but the authors are asked to provide additional information and to re-frame this retrospective analysis as a pilot study evaluating different approaches to determining disease severity. Validation is a bit too definitive.

1.The study population is not fully defined. For clinical purposes, are patients classified as either ME or CFS and then treated/managed the same? From line 83 it is explained that the studied patient group fulfilled ALL the ME and CFS criteria. It was made more clear on line 83.

Of the 714 presenting as presumed ME/CFS, what were the reasons that 39 were not considered to be ME/CFS (symptoms, exclusionary conditions, alternative diagnoses, other)? added

Other parameters of the complex ME/CFS phenotype are omitted, such as pain, sleep, cognition, orthostatic intolerance, PEM, medication use, sudden vs gradual onset, comorbidities use as fibromyalgia, etc. are not described. This information may not be available in retrospective review, but this kind of heterogeneity deserves comment and should be included as a limitation. We respectfully disagree with the reviewer. This paper is not on types of symptoms, not on onset but on validating clinical criteria as suggested by Carruthers et al in 2011 with a regularly used questionnaire and two more objective measures. Heterogeneity is present but allocating severity is suggested by the ICC irrespective of heterogeneity of symptoms. Heterogeneity might influence the spread of information on the measures though. This has been added to the limitation section.

  1. How do the patients with valid assessments for all 3 measures within 3 months compare with those who could not be included in the study? Of 675 patients with ME/CFS less than half [only 289 (40.5%)] are in the study. This could introduce significant bias in the assessment. No difference was found in 75 patients with a longer than 3 month interval between the measures. This information has been added in new lines 90-91. We have also added information to the limitation section.

  2. The inter-observer agreement for the disease severity classification of patients using the ICC criteria appears to quite good (kappa = 0.86), however this was performed on only slightly more than half of the study participants (162/289, 56%) and the second clinician made the determination through record rather than during clinic visit. Was the clinic physician’s determination of disease severity included in patient charts, if so, how was this redacted to allow unbiased determination. What is the reliability of clinical record determination of disease severity compared with determinations made at clinical visits? The second clinician was blinded for severity comments by the first clinician and only worked with history of the patient as presented in the clinical record. The conclusion section with the severity grading was blinded. With the fairly good agreement we did not explore the reasons for a difference in opinion per case.

  3. The CPET is described as symptom limited and different rates of ramping are used. This does not seem to follow usual practice, where determinations are made using either maximal or submaximal challenge. This form of CPET could be useful, but further description and justification would be helpful. CPET can be stressful to patients with ME/CFS. It would be helpful for the authors to comment on any adverse outcomes associated with CPET. For exercise intolerance measurements, tests in our specialty clinic are always maximal test. Using different ramping is necessary due to the differences in age, gender and information on exercise tolerance as obtained from the detailed history (as is stated on new line 120). Except for eliciting post-exertional malaise, which is an obvious result of exercising either maximal or submaximal in ME/CFS, no other adverse events were reported. We added the fact it was a maximal test at new line 118.

  4. The Sensewear activity armband step count was normalized to 24 hours. Why wasn’t the step count used directly? What is gained by normalizing to 24 hours? The information was normalized to 24 hours to account for information of steps from only part of a day. This was added on new line 104

  5. The ICC proposal for disease severity includes 4 categories, but the report describes only 3. The very severe category may not be represented in the clinic population. In addition, their illness may preclude having all three measures, particularly the CPET. It would be helpful if this could be addressed by the authors. Very severe patients never undergo a CPET. We added the information on new lines 81-82

  6. Results shown in Figure 1 as a bar graph may be more informative if shown as box plots to give a clearer idea of the distribution and numbers. Data shown in Figure 2 may be more helpful shown in table form with sensitivity and specificity for each classification. The ROC data may be clearer if presented graphically on sensitivity/specificity analysis. To avoid flooding the manuscript with numbers, we performed the ROC analysis as described by Liu and determined the cut-off values, taking the clinical grading in this case as “gold” standard: we showed the cut-off values with the highest number resulting from multiplying sensitivity We added additional information. We changed Figure 1 from columns to box plots as requested. Figure 2 data are clarified in a table now, so the text describing this figure has been deleted. In the physical activity subscale of the RAND for mild patients, the mean bar is the same as the lower border of the box plot. It looks missing, but this is a correct version of the results.

  7. Discussion (section 4.4) presents information not on the study population not included in the results. The ambiguous use of “this study” should be avoided as it is confusing whether “this study” refers to the study being presented or the one referenced. What is being conveyed by the statement “Both studies found high mean absolute % error for Sensewear armband relative to other activity trackers”? This seems to suggest Sensewear may not be valid or reliable and some clarification is needed. We checked section 4 for this point and updated where appropriate. The comments in the Discussion’s other sections with possible confusion were clarified. We added additional sentences on the topic of the mean absolute error to clarify.
  8. Additional limitations should be included. Only one clinical practice is evaluated. The data could be used to focus a prospective analysis of the most promising measures in multiple clinical settings. added